# Rapid detection of extended spectrum β-lactamase producing *Escherichia coli* isolated from fresh pork meat and pig cecum samples using multiplex recombinase polymerase amplification and lateral flow strip analysis

**Sitthichai Kanokudom**[1], **Thachaporn Assawakongkarat**[1☯], **Yukihiro Akeda**[2,3☯], **Panan Ratthawongjirakul**[1☯], **Rungtip Chuanchuen**[4☯], **Nuntaree Chaichanawongsaroj**[1]*

**1** Research Unit of Innovative Diagnosis of Antimicrobial Resistance, Department of Transfusion Medicine and Clinical Microbiology, Faculty of Allied Health Sciences, Chulalongkorn University, Bangkok, Thailand, **2** Division of Infection Control and Prevention, Osaka University Hospital, Osaka University, Suita, Japan, **3** Research Institute for Microbial Diseases, Osaka University, Suita, Japan, **4** Research Unit in Microbial Food Safety and Antimicrobial Resistance, Department of Veterinary Public Health, Faculty of Veterinary Science, Chulalongkorn University, Bangkok, Thailand

☯ These authors contributed equally to this work.

* nuntaree@gmail.com

**Data Availability Statement:** All relevant data are within the paper.

## Abstract

The emergence and dissemination of extended-spectrum β-lactamase (ESBL)-producing *Escherichia coli* is a global health issue. Food-producing animals, including pigs, are significant reservoirs of antimicrobial resistance (AMR), which can be transmitted to humans. Thus, the rapid detection of ESBLs is required for efficient epidemiological control and treatment. In this study, multiplex recombinase polymerase amplification (RPA) combined with a single-stranded tag hybridization chromatographic printed-array strip (STH-PAS), as a lateral flow strip assay (LFA), was established for the rapid and simultaneous detection of multiple *bla* genes in a single reaction. Visible blue lines, indicating the presence of the *bla*$_{CTX-M}$, *bla*$_{SHV}$, and *bla*$_{OXA}$ genes, were observed within 10 min by the naked eye. The limit of detection of all three genes was 2.5 ng/25 μL, and no cross-reactivity with seven commensal aerobic bacteria was observed. A total of 93.9% (92/98) and 96% (48/50) of the *E. coli* isolates from pork meat and fecal samples, respectively, expressed an ESBL-producing phenotype. Nucleotide sequencing of the PCR amplicons showed that *bla*$_{CTX-M}$ was the most prevalent type (91.3–95.83%), of which the main form was *bla*$_{CTX-M-55}$. The sensitivity and specificity of the RPA-LFA were 99.2% and 100%, respectively, and were in almost perfect agreement (κ = 0.949–1.000) with the results from PCR sequencing. Thus, the RPA-LFA is a promising tool for rapid and equipment-free ESBL detection and may facilitate clinical diagnosis in human and veterinary medicine, as well as AMR monitoring and surveillance.

**Funding:** This research project was granted by Agricultural Research Development Agency (Public Organization) or "ARDA" and partially supported by Research Unit of Innovative Diagnosis of Antimicrobial Resistance, Ratchadapisek Sompoch Endowment fund and the Second Century Fund (C2F), Chulalongkorn University and Japan Agency for Medical Research and Development (AMED). We would like to extend our sincerest gratitude to Graduate School as well as Faculty of Allied Health Sciences at Chulalongkorn University for providing the Overseas Research Experience Scholarship for Graduate Students. The funders had no role in study design, data collection and analysis, decision to publish, or preparation of the manuscript.

**Competing interests:** The authors have declared that no competing interests exist.

## Introduction

The rising level of antimicrobial resistance (AMR) worldwide has a significant impact on humans, animals, and the environment. One of the major reservoirs of AMR organisms and determinants is food animals, especially pigs and poultry, due to the overuse/misuse of antibiotics in farms to prevent and reduce the risk of infection [1]. Apart from antibiotic residues in food, antibiotic resistance genes can disseminate from animals to animals and animals to humans via the food chain [2]. *Escherichia coli*, a part of the commensal flora in human and animal intestines, acts as a fecal index of food hygiene, and simultaneously represents the carriage of AMR [3–6].

Extended spectrum β-lactamase (ESBL)-producing *E. coli* have been categorized by the World Health Organization (WHO) as being the most critical AMR pathogen to human health and a major public health concern. The incidence of ESBL-producing *E. coli* has increased globally in hospitals and livestock [7–9]. Broad-spectrum β-lactamase enzymes hydrolyze penicillins, cephalosporins (first-, second- and third-generation), and aztreonam but not carbapenems. However, these enzymes are inhibited by clavulanate [10]. In Thailand, a previous study has reported that the prevalence of ESBL-producing *E. coli* in the food industries and in animal farmworkers, pigs, fresh pig meat (pork), and water sources from pig farms ranges from 61.5–77.3%, which is higher than that in fresh vegetables, fish and cooked foods, as well as in canals and shrimp ponds [11]. Up to 86% of bacterial isolates (97.8% *E. coli*) from food animals, including pigs, cattle, chicken, and sheep, produce CTX-M group 1 [2]. Recently, up to 92% of CTX-M groups have been detected among the multidrug-resistant (MDR) *E. coli* isolates from healthy finisher and breeder pigs and boot swabs from pig farms [12].

According to the Clinical and Laboratory Standards Institute (CLSI), phenotypic screening and confirmation is the standard method for ESBL detection. However, the method is laborious, time-consuming, and unable to discriminate between different ESBL genotypes. In addition, the low expression level and coexistence of more than one AMR gene, especially in different drug classes, may confer false susceptibility or resistance results [13, 14]. Several molecular techniques have been used to detect the presence of ESBL genes, including PCR, real-time PCR, microarray, and next-generation sequencing [15–17], but most of these techniques require expensive instruments and well-trained personnel to operate and analyze the results.

Recombinase polymerase amplification (RPA), first introduced in 2006, has been applied in various fields, particularly for the diagnosis of infectious diseases. RPA is simple to perform and uses a single low temperature of 37–42°C with two primers. The double-stranded DNA template is dissociated by a recombinase and single-stranded binding protein, and then, *Sau* DNA polymerase generates new complementary products within 5–20 min [18]. The results of alternative post-RPA detection methods using SYBR Green or a lateral flow strip assay (LFA) are easily observed by the naked eye, and can be applied for point-of-care testing and field studies [19–21]. Hence, this study attempted to develop multiplex RPA combined with LFA to detect the three most common ESBL genes ($bla_{CTX-M}$, $bla_{OXA}$, and $bla_{SHV}$). ESBL-producing *E. coli* was obtained from pork collected from retailed markets in Bangkok, Thailand, and pig cecum samples from a wholesale slaughterhouse. Each antibiotic susceptibility profile was examined by disk diffusion, and ESBL variants were characterized by nucleotide sequencing and the RPA-LFA.

## Materials and methods

### Sample collection

A total of 100 pork samples (200 g each) were randomly purchased from one vendor each from 100 fresh markets in Bangkok, Thailand, from June to December 2019. Samples were

collected, kept on ice, and transported to the laboratory for analysis within 24 h. Fifty pig cecum samples were obtained from a wholesale slaughterhouse in Suphanburi Province, Thailand, in February 2020. The history of antibiotic administration to the pigs was unknown. All samples were kept at 4˚C until processed.

## Isolation of ESBL-producing *E. coli*

Each pork meat sample was gently chopped into small pieces, and then, 25 g of chopped pork was added to 225 mL of sterile peptone water and incubated at 37˚C overnight. One loop of the peptone culture was streaked onto MacConkey agar (Oxoid, UK) supplemented with 1 g/L cefotaxime (Oxoid, UK). Fecal material from cecum samples from each pig was collected using a sterile cotton swab and directly inoculated on MacConkey agar supplemented with 1 g/L cefotaxime. Three typical lactose fermented (LF) colonies were randomly selected from each sample, screened for *E. coli* on eosin methylene blue agar (Oxoid, UK), and identified by conventional biochemical testing, including MIO, TSI, LIA, citrate, and urea utilization [22]. If all the isolates from the same sample were designated as *E. coli* by the same biochemical results, then a single isolate was randomly selected.

All the selected *E. coli* isolates were subsequently confirmed by PCR amplification of *UspA* using *E. coli*-specific primers (Table 1). PCR included 0.05 μM of each primer, 200 μM dNTPs, 0.625 U Taq polymerase (NEB, Inc., USA), and 50 ng DNA template in a total volume of 25 μL. The PCR amplification conditions were as follows: 94˚C for 5 min, 35 cycles of 94˚C for 30 sec, 60˚C for 15 sec, 72˚C for 30 sec, and 72˚C for 5 min.

## Antimicrobial susceptibility testing

The susceptibility to 21 antimicrobial agents (ampicillin, cefpodoxime, ceftriaxone, cefepime, cefotaxime, ceftazidime, aztreonam, amoxicillin-clavulanate, piperacillin-tazobactam, imipenem, meropenem, amikacin, gentamycin, kanamycin, streptomycin, ciprofloxacin, nalidixic acid, tetracycline, trimethoprim-sulfamethoxazole, azithromycin and chloramphenicol) was determined using the disk diffusion method. The production of ESBL was detected by the

**Table 1. Primer sequences and their product sizes.**

| Target gene | Primer sequence (5' to 3') | Amplicon size (bp) | Reference |
|---|---|---|---|
| *uspA* | F–AATGCAGGCTACCCAATCAC | 162 | This study |
| | R–GGTGTTGATCAGCTGACGTG | | |
| *bla*CTX-M | F–ATGTGCAGYACCAGTAARGTKATGGC | 593 | [23] |
| | R–TGGGTRAARTARGTSACCAGAAYCAGCGG | | |
| *bla*TEM | F–TCCGCTCATGAGACAATAACC | 931 | [24] |
| | R–TTGGTCTGACAGTTACCAATGC | | |
| *bla*SHV | F–TGGTTATGCGTTATATTCGCCC | 868 | [25] |
| | R–GGTTAGCGTTGCCAGTGCT | | |
| *bla*OXA | F–ACACAATACATATCAACTTCGC | 814 | [26] |
| | R–AGTGTGTGTTTAGAATGGTGATC | | |
| *bla*CTX-M-tag | F–Biotin–ATGTGCAGYACCAGTAARGTKATGGC | 593 | This study |
| | R–tag1–TGGGTRAARTARGTSACCAGAAYCAGCGG | | |
| *bla*SHV-tag | F–tag2–GATGAACGCTTTCCCATGATG | 214 | This study |
| | R–Biotin–CGCTGTTATCGCTCATGGTAA | | |
| *bla*OXA-tag | F–Biotin–ATTATCTACAGCAGCGCCAGTG | 296 | This study |
| | R–tag3–TGCATCCACGTCTTTGGTG | | |

combination disk method as recommended by CLSI [27]. *E. coli* ATCC 25922 was used as a quality control strain.

## Characterization of ESBL genes by PCR-sequencing

Four common ESBL genes ($bla_{CTX-M}$, $bla_{OXA}$, $bla_{SHV}$, and $bla_{TEM}$) were screened in all the bacterial isolates by PCR and confirmed by DNA sequencing. Each obtained sequence was subsequently compared to known genotype sequences in the NCBI database using the BLASTn program. In brief, whole-cell DNA from the bacterial isolates was prepared by the boiling method. PCR was subsequently performed using gene-specific primers (Table 1). Each PCR was composed of 0.2 μM of each primer, 200 μM dNTPs, 0.625 U Taq polymerase (NEB Inc., USA), and 50 ng DNA template in a total volume of 25 μL. PCR amplification was performed at an initial temperature of 94˚C for 5 min followed by 35 cycles of 94˚C for 30 sec, 60˚C for 15 sec, and 72˚C for 30 sec and then a final 72˚C for 5 min. The PCR products were analyzed by 1.5% (w/v) agarose gel electrophoresis and submitted for commercial nucleotide sequencing analysis (Bioneer Corporation, South Korea).

## Multiplex RPA-LFA

A TwistDx basic kit (TwistDx, UK) was used for the amplification of the $bla_{CTX-M}$, $bla_{OXA}$, and $bla_{SHV}$ genes according to the manufacturer's instructions. The sequences of the specific tag-primers used in the RPA reactions are shown in Table 1. The positive DNA controls, *E. coli* EC 137 (harboring $bla_{CTX-M}$ and $bla_{OXA}$) and *Klebsiella pneumoniae* KP 125 (harboring $bla_{SHV}$), were kindly provided by Prof. Dr. Visanu Thamlikitkul, Faculty of Medicine Siriraj Hospital, Mahidol University, Bangkok, Thailand. In brief, the RPA-LFA conditions were optimized for a uniplex reaction (0.2 μM $bla_{SHV}$ tag primers) and multiplex reactions of two (0.24 μM of $bla_{CTX-M}$ and 0.06 μM $bla_{OXA}$ tag primers) and three primer sets (0.2 μM $bla_{SHV}$, 0.2 μM of $bla_{CTX-M}$ and 0.1 μM $bla_{OXA}$ tag primers). Each primer set was mixed with 29.5 μL rehydration buffer, a freeze-dried protein pellet, 1 μL of 50 ng DNA template, and 1.25 μL of 280 mM magnesium acetate in a final reaction volume of 25 μL. The reaction was incubated at 37˚C for 30 min in a heat-dried bath. The RPA amplicons were purified using a NucleoSpin gel and PCR clean-up kit (Macherey-Nagel GmbH & Co., Germany).

The purified RPA products (uniplex, duplex and triplex RPA) were detected using a single-stranded tag hybridization chromatographic printed-array strip (STH-PAS; TBA Co., Ltd., Japan). Briefly, 10 μL of 300 mM NaCl-modified dilution buffer and 1 μL of streptavidin-coated blue latex were added to 10 μL of the RPA product. The C-PAS4 membrane stick was dipped into the mixture for 5–10 min. The uniplex:duplex RPA products were evaluated at ratios (v/v) of 1:4, 2:4, 3:4 and 4:4. Each visible blue line indicated the presence of the respective ESBL target gene, which were from the bottom to top lines: $bla_{CTX-M}$, $bla_{SHV}$, and $bla_{OXA}$ (Fig 1).

The specificity of the RPA-LFA was ascertained using *E. coli* ATCC 25922, *Klebsiella pneumoniae* ATCC 700603, *Acinetobacter baumannii* ATCC 19606, *Pseudomonas aeruginosa* ATCC 27853, *Proteus mirabilis* ATCC 25933, *Staphylococcus aureus* ATCC 25923, and *Enterococcus faecalis* ATCC 29212. The limit of detection (LOD) for each ESBL gene was examined using various concentrations of each positive DNA template (0, 0.1, 1, 2.5, 5, 10, 50, and 100 ng/25 μL).

## Detection of the $bla_{CTX-M}$, $bla_{OXA}$, and $bla_{SHV}$ genes in ESBL-producing *E. coli* by the RPA-LFA

All the ESBL-positive *E. coli* isolates from the pork, and cecum samples were screened for the presence of $bla_{CTX-M}$, $bla_{OXA}$, and $bla_{SHV}$ by the RPA-LFA. The TwistDx basic kit and STH-PAS dipstick were employed using the conditions determined in the present study.

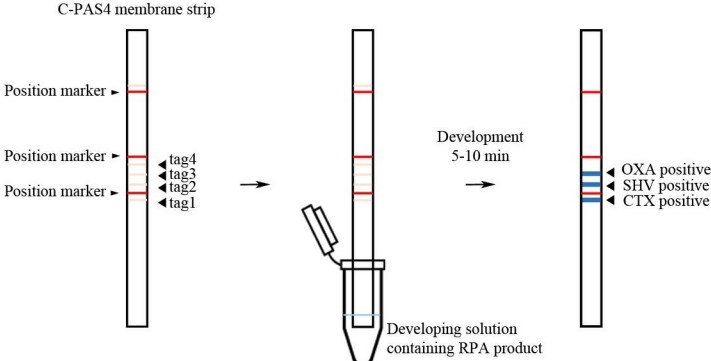

**Fig 1. Schematic illustration of the RPA-LFA for the detection of $bla_{CTX-M}$, $bla_{OXA}$, and $bla_{SHV}$.** The C-PAS4 membrane strip has 3 position markers (red lines) and four imprinted tag oligonucleotides from the bottom to top lines (tag1-4; Left). The C-PAS4 membrane strip is inserted into a developing solution containing RPA amplicons (middle) for 5–10 min to allow the RPA amplicons to be trapped by the complementary tag sequences and generate visible blue lines (CTX, SHV, OXA) at the relevant positions, as indicated (right).

## Data analysis

The results of the RPA-LFA and PCR detection were compared, and the sensitivity and specificity of the RPA-LFA were analyzed using Medcalc® software (https://www.medcalc.org/calc/diagnostic_test.php). The level of agreement between the two diagnostic methods was evaluated for significance using kappa (κ) statistics with QuickCals software (https://www.graphpad.com/quickcalcs/kappa1) [28].

## Results

### Antimicrobial susceptibility patterns of *E. coli* isolated from retail pork meat and pig cecum samples

From the 100 pork samples individually collected from 100 retail markets, a single *E. coli* isolate was selected from 68 positive samples. Two and three isolates designated as *E. coli* with different biochemical results were selected from 9 and 4 samples, respectively. Thus, a total of 98 *E. coli* isolates were obtained from 81 pork samples. Of these screened *E. coli* isolates, 93.9% (92/98) were ESBL producers, and three isolates were resistant to carbapenem (Fig 2A).

From the fecal samples, a single *E. coli* isolate was selected from each of the 50 fecal samples collected from the slaughterhouse, and 96% (48/50) were ESBL producers (Fig 2C). Almost all the ESBL-producing isolates (92.14%) were found to display MDR, and they were resistant to at least one agent in three or more antimicrobial categories. The ESBL-MDR isolates remarkably resisted trimethoprim-sulfamethoxazole, gentamycin, tetracycline, and chloramphenicol (Fig 2B and 2D). Notably, a markedly higher percentage of ESBL-MDR isolates were found in the pig cecum samples (87.5–100%) than in the pork samples (45.6–88.0%).

### Genotyping of ESBL-producing isolates

Among the total ESBL-positive-isolates, 91.3% (84/92) and 95.83% (46/48) carried $bla_{CTX-M}$ genes in the pork and cecum samples, respectively, either alone or in combination with $bla_{TEM-1}$. Others isolates included $bla_{SHV-12, TEM-1}$ (n = 1), $bla_{SHV-12}$ (n = 1), $bla_{TEM-1}$ (n = 7) and other ESBL gene/AMR mechanisms (n = 1). CTX-M group 9 type 55 was predominant (50%, 46/92) in the pork and pig cecum samples (66.7%, 32/48) (Fig 3). The $bla_{CTX-M}$ variants in the pork samples derived from the 100 individual markets were diverse and comprised 14 CTX-M

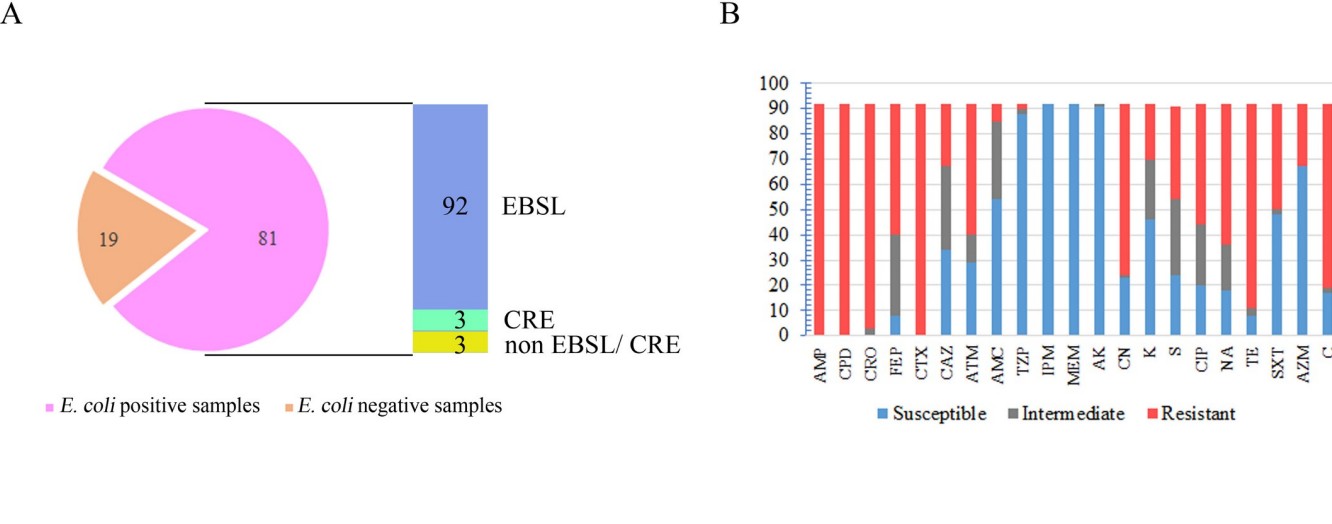

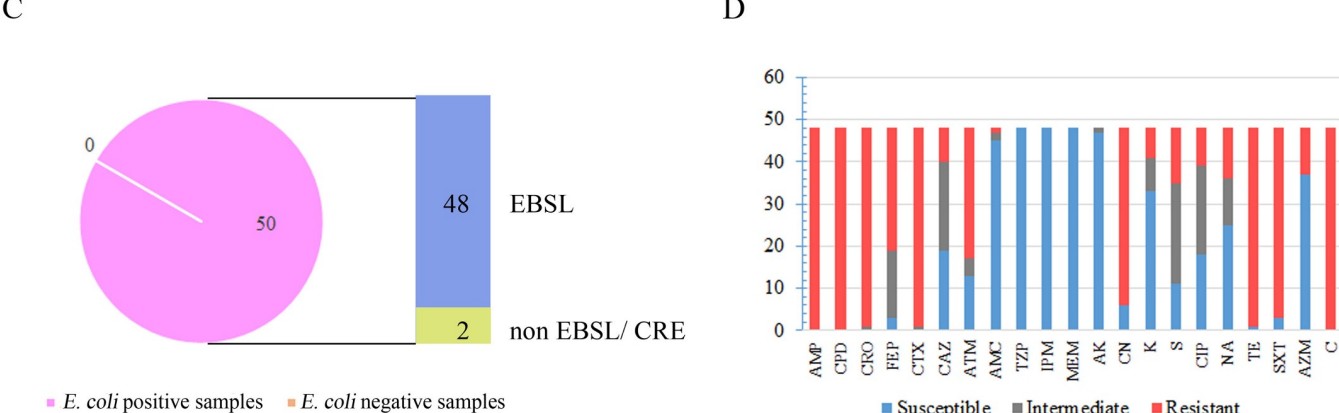

**Fig 2. Prevalence of ESBL-producing *E. coli* isolated from fresh pork meat and pig cecum samples and their antibiotic susceptibility profiles.** Numbers of (A, C) ESBL-producing *E. coli* and their (B, D) antibiotic-resistant profiles ascertained by disk diffusion, from (A, B) pork and (C, D) pig cecum samples. CRE; carbapenemase resistant *Enterobacteriaceae*.

types belonging to CTX-M groups 1 and 9 (CTX-M-1 group: CTX-M-15, CTX-M-55, CTX-M-79, and CTX-M-136; CTX-M-9 group: CTX-M-9, CTX-M-14, CTX-M-16, CTX-M-24, CTX-M-27, CTX-M-65, CTX-M-112, and CTX-M-161) and unclassified groups (CTX-M-176 and CTX-M-227). However, only four CTX-M types (CTX-M-55, CTX-M-14, CTX-M-15, and CTX-M-27) were observed in the cecum samples (Table 2).

## Optimization of the RPA-LFA for the detection of $bla_{CTX-M}$, $bla_{OXA}$, and $bla_{SHV}$

The uniplex RPA-LFA successfully amplified $bla_{CTX-M}$, $bla_{OXA}$, and $bla_{SHV}$, each of which showed a visible blue line at the designated position on the dipstick strip (an example of $bla_{SHV}$ is shown in Fig 4A, no. 1 and 2). However, the multiplex RPA-LFA with all three primer sets resulted in a false-positive line for the $bla_{SHV}$ gene in the no DNA template negative control (Fig 4A, no. 5 and 6). The duplex $bla_{CTX-M}$ and $bla_{OXA}$ primers yielded positive results at the designated lines (Fig 4A, no. 3 and 4) with no cross-reactivity in the negative control. The optimum RPA-LFA condition for $bla_{CTX-M}$, $bla_{OXA}$, and $bla_{SHV}$ detection (Fig 4B) was determined by

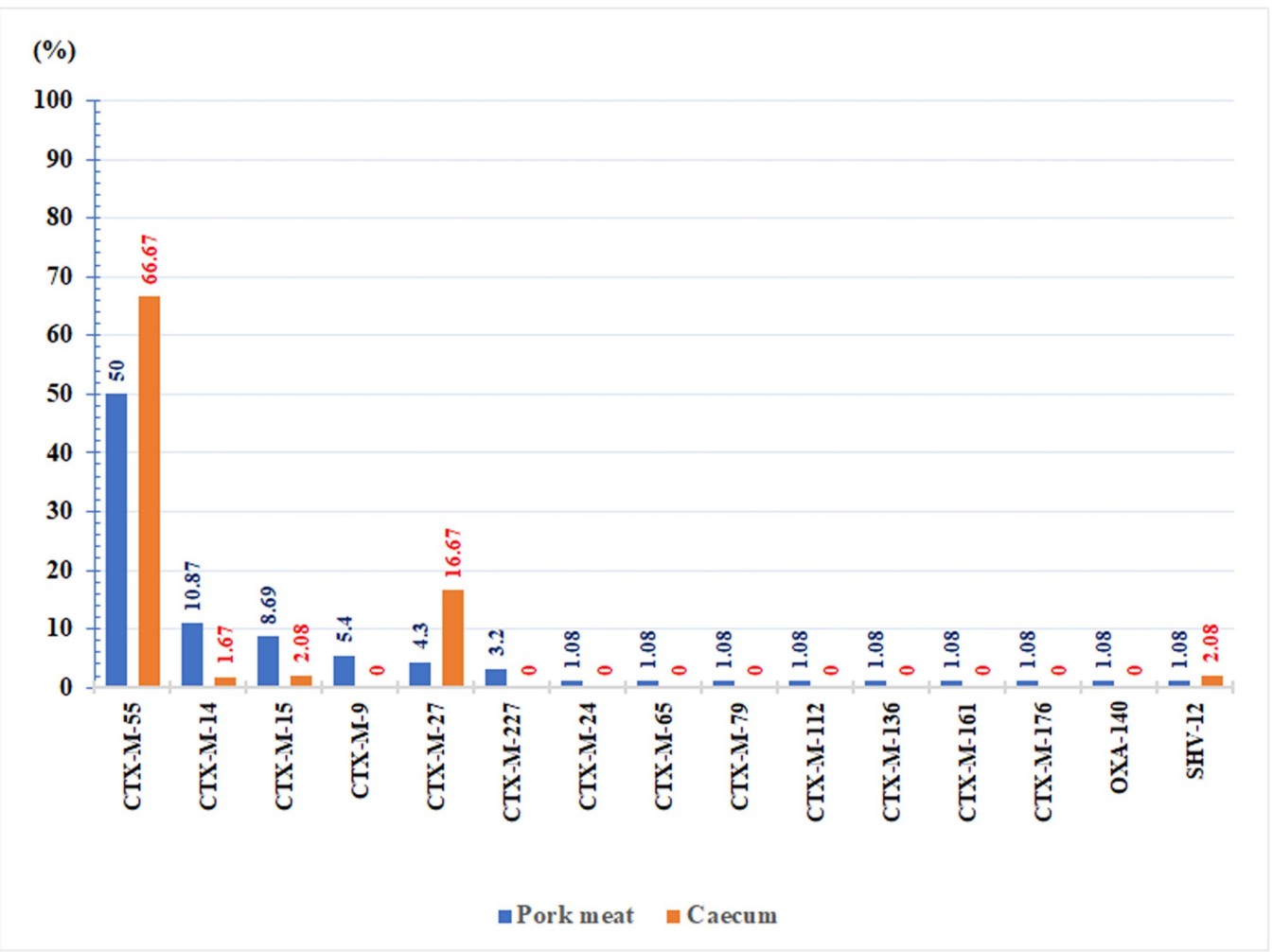

**Fig 3. Percentage of ESBL subtypes in ESBL-producing *E. coli* from pork (n = 92) and pig cecum samples (n = 48).**

mixing the uniplex RPA product of $bla_{SHV}$ with the duplex RPA products of $bla_{CTX-M}$ and $bla_{OXA}$ at a 2:4 (v/v) ratio before LFA detection. The $bla_{TEM}$ gene was not detectable by RPA due to $bla_{TEM}$ contamination in the engineered recombinase enzyme in the TwistDx kit, which originated from $bla_{TEM-1}$-containing *E. coli* K-12 and resulted in a false positive TEM result [29, 30].

## Specificity of the RPA-LFA for the detection of $bla_{CTX-M}$, $bla_{OXA}$, and $bla_{SHV}$

As shown in Fig 5, *K. pneumoniae* ATCC 700603, carrying the $bla_{SHV}$ gene, and the three ESBL-positive DNA templates ($bla_{CTX-M}$, $bla_{OXA}$, and $bla_{SHV}$) showed positive lines at their respective positions on the strip. No cross-reaction was detected among the $bla_{CTX-M}$, $bla_{OXA}$, and $bla_{SHV}$ genes in *E. coli* ATCC 25922, *K. pneumoniae* ATCC 700603, *A. baumannii* ATCC 19606, *P. aeruginosa* ATCC 27853, *P. mirabilis* ATCC 25933, *S. aureus* ATCC 25923, and *E. faecalis* ATCC 29212.

## LOD of the RPA-LFA $bla_{CTX-M}$, $bla_{OXA}$, and $bla_{SHV}$

To measure the sensitivity of the RPA-LFA for ESBL gene detection, the $bla_{CTX-M}$, $bla_{OXA}$, and $bla_{SHV}$ DNA templates were screened at 0, 0.1, 2.5, 5, 10, 50, and 100 ng/25 μL. Positive results

**Table 2. Presence of the *bla*~CTX-M~, *bla*~TEM~, *bla*~OXA~, and *bla*~SHV~ genes in ESBL-producing *E. coli* from fresh pork and cecum samples (n = 140).**

| Beta-lactamase genes in *E. coli* | Fresh pork meat | Pig cecum |
|---|---|---|
| | No (%) | No (%) |
| | Total (n = 92) | Total (n = 48) |
| CTX-M-55, TEM-1 | 28 (30.4) | 23 (47.9) |
| CTX-M-55 | 15 (16.3) | 9 (18.8) |
| CTX-M-55, TEM-30 | 1 (1.08) | - |
| CTX-M-55, OXA-140 | 1 (1.08) | - |
| CTX-M-55, TEM-176 | 1 (1.08) | - |
| CTX-M-14 | 5 (5.43) | 2 (4.16) |
| CTX-M-14, TEM-1 | 5 (5.43) | 3 (6.25) |
| CTX-M-15 | 3 (3.24) | - |
| CTX-M-15, TEM-1 | 4 (4.32) | 1 (2.08) |
| CTX-M-15, OXA-1 | 1 (1.08) | - |
| CTX-M-9 | 3 (3.24) | - |
| CTX-M-9, TEM-1 | 1 (1.08) | - |
| CTX-M-9, TEM-176 | 1 (1.08) | - |
| CTX-M-27 | 2 (2.16) | 1 (2.08) |
| CTX-M-27, TEM-1 | 2 (2.16) | 7 (14.6) |
| CTX-M-227 | 1 (1.08) | - |
| CTX-M-227, TEM-1 | 2 (2.16) | - |
| CTX-M-16, TEM-1 | 1 (1.08) | - |
| CTX-M-24 | 1 (1.08) | - |
| CTX-M-65 | 1 (1.08) | - |
| CTX-M-79 | 1 (1.08) | - |
| CTX-M-112 | 1 (1.08) | - |
| CTX-M-136 | 1 (1.08) | - |
| CTX-M-161, TEM-1 | 1 (1.08) | - |
| CTX-M-176 | 1 (1.08) | - |
| SHV-12 | - | 1 (2.08) |
| SHV-12, TEM-1 | 1 (1.08) | - |
| TEM-1 | 6 (6.52) | 1 (2.08) |
| Other ESBL genes/AMR mechanisms | 1 (1.08) | - |

were determined as obvious blue lines. The LOD for all three genes was found to be 2.5 ng/ 25 µL (Fig 6).

## Comparison of the RPA-LFA and PCR for the detection of the *bla*~CTX-M~, *bla*~OXA~, and *bla*~SHV~ genes

To examine the efficiency of the RPA-LFA, a total of 140 ESBL-producing *E. coli* isolates (92 ESBL-positive pork isolates and 48 ESBL-positive fecal isolates) were tested by the RPA-LFA developed in the present study, and the results were compared to the results obtained by screening the same isolates by diagnostic PCR. Of 130 *bla*~CTX-M~ PCR-positive samples, a discrepancy was revealed in one of the *bla*~CTX-M~, *bla*~TEM~ samples by RPA-LFA (Table 3). According to agreement analysis, both methods showed completely matched results, with 129 positive and 10 negative *bla*~CTX-M~ samples and two positive and 138 negative *bla*~SHV~ and *bla*~OXA~ samples, except one false-positive result (compared to the PCR) detected for *bla*~CTX-M~ by the RPA-LFA (Tables 3 and 4). The otherwise concordant results of both methods provided

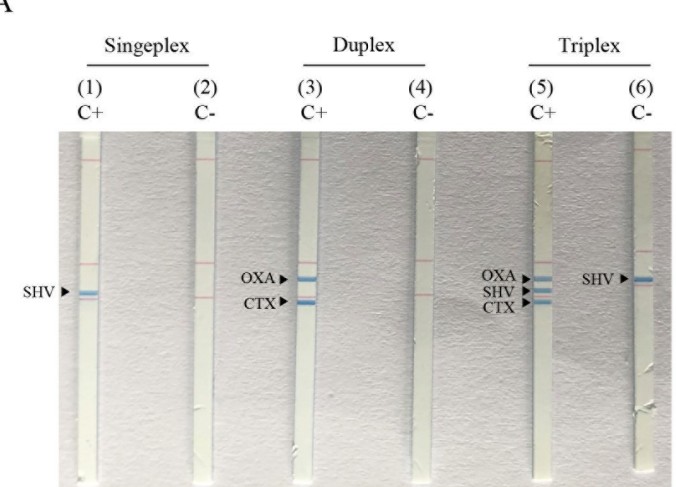

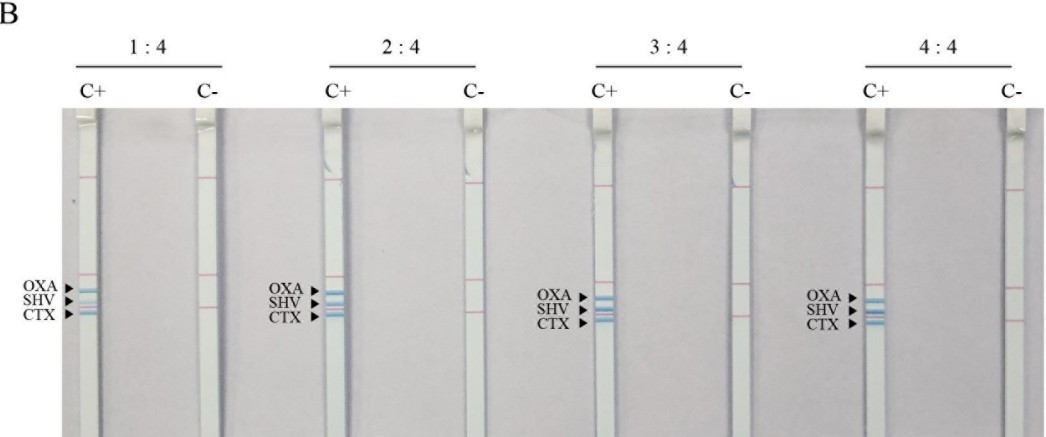

**Fig 4. Detection of the *bla*CTX-M, *bla*OXA, and *bla*SHV genes by the RPA-LFA.** A visible blue line represents a positive detection. Detection of uniplex (no.1 and 2), duplex (no. 3 and 4), and triplex *bla* genes (no. 5 and 6) from the three *bla* genes (A). The combination of the uniplex *bla*SHV product with the duplex *bla*CTX-M and *bla*OXA products at 1:4, 2:4, 3:4 and 4:4 (v/v) ratios (B). C+; positive DNA control, C-; no template control.

almost perfect agreement ($k$ = 0.949 for *bla*CTX-M and $k$ = 1.000 for both *bla*SHV and *bla*OXA). The sensitivity of the RPA-LFA for *bla*CTX-M and both *bla*SHV and *bla*OXA detection were 99.2% (95% confidence interval [CI] = 95.8–100%) and 100% (95% CI = 15.8–100%), respectively. The specificity of the RPA-LFA for *bla*CTX-M and both *bla*SHV and *bla*OXA detection were 100% (95% CI = 69.2–100%) and 100% (95% CI = 97.4–100%), respectively.

## Discussion

Due to some of the limitations of phenotypic detection methods and the nonportability, expense, and technical skill required for the available advanced genotypic assays, the development of a more rapid and equipment-free system for the detection of ESBL genes is required. The aim of this study was to develop a rapid and simple diagnostic test for the most common ESBL genes encoding CTX-M, OXA, and SHV using an RPA-LFA method. The developed

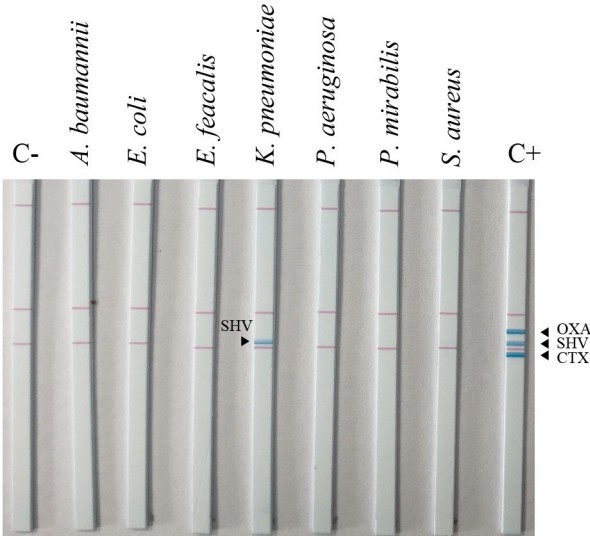

**Fig 5. Specificity of the RPA-LFA for *bla*$_{CTX-M}$, *bla*$_{OXA}$, and *bla*$_{SHV}$.** C+, positive DNA control; C-, no template control.

RPA could rapidly amplify *bla*$_{CTX-M}$, *bla*$_{OXA}$, and *bla*$_{SHV}$ at 37°C in 30 min with easily interpreted visual results by the naked eye on lateral flow strips. Although, our RPA-LFA utilized two separate duplex (*bla*$_{CTX-M}$, *bla*$_{OXA}$) and uniplex (*bla*$_{SHV}$) RPA reactions, the amplification was simple and rapid with no need for an expensive thermal cycler. Moreover, there is no need for gel preparation and staining, which shortens the detection time and decreases the amount of cumbersome processing. Multiplex isothermal amplifications are different among different strategies. Detection by multiplex loop-mediated isothermal amplification is difficult to perform because of the complicated structure of the primers and the recommended amplicon size of < 250 bp. Only a few multiplex helicase-dependent amplifications have been reported because the short amplicon size of < 150 bp is a limitation. Likewise, strand displacement amplification requires at least four primers/target and restriction enzyme digestion, which obstructs multiplexing [31]. The benefits of RPA are its low operating temperature (37–42°C),

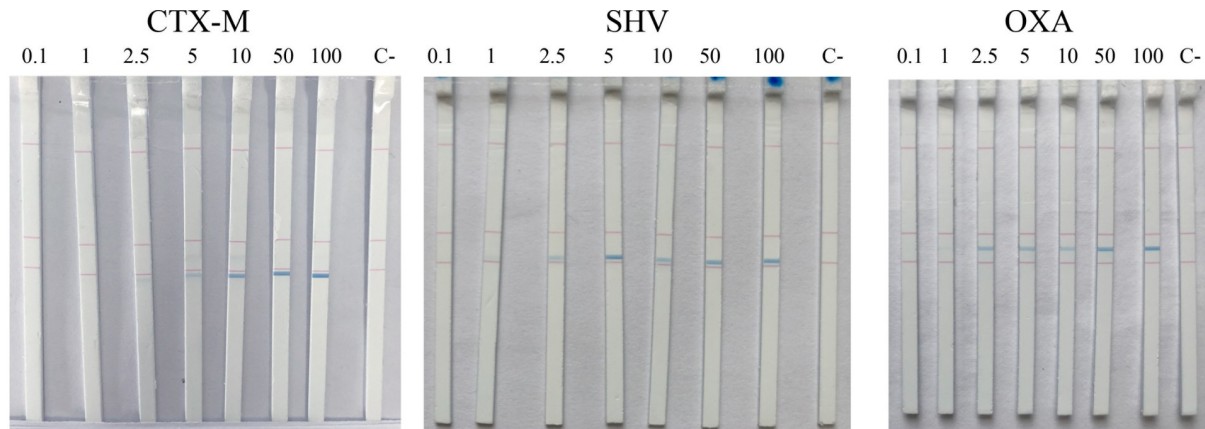

**Fig 6. LODs for *bla*$_{CTX-M}$, *bla*$_{OXA}$, and *bla*$_{SHV}$ detection of the RPA-LFA.** The LODs for *bla*$_{CTX-M}$, *bla*$_{OXA}$, and *bla*$_{SHV}$ of the RPA-LFA method were determined using various concentrations (0, 0.1, 2.5, 5, 10, 50 and 100 ng/25 μL) of the respective DNA template. C-, no template control.

**Table 3. Pairwise comparison between RPA-LFA and PCR detection of $bla_{CTX-M}$, $bla_{OXA}$, and $bla_{SHV}$ (n = 140).**

| Beta lactamase genes | PCR | RPA-LFA | | |
|---|---|---|---|---|
| | | $bla_{CTX-M}$ | $bla_{OXA}$ | $bla_{SHV}$ |
| $bla_{CTX-M}$ | 130 | 129 | 2 | 0 |
| $bla_{CTX-M}$ alone | 47 | 47 | 0 | 0 |
| $bla_{CTX-M}$, $bla_{TEM}$ | 81 | 80 | 0 | 0 |
| $bla_{CTX-M}$, $bla_{OXA}$ | 2 | 2 | 2 | 0 |
| $bla_{SHV}$ | 2 | 0 | 0 | 2 |
| $bla_{SHV}$ alone | 1 | 0 | 0 | 1 |
| $bla_{SHV}$, $bla_{TEM}$ | 1 | 0 | 0 | 1 |
| $bla_{TEM}$ | | | | |
| $bla_{TEM}$ alone | 7 | 0 | 0 | 0 |
| Other ESBL genes/AMR mechanisms | 1 | 0 | 0 | 0 |

which is compatible with general heating instruments in the laboratory or even body temperature; its lack of requirement for tight temperature control and initial denaturation; and its rapid amplification time of within 5–25 min. The recombinase typically consumes all the available ATP within 25 min [32]. Although, our multiplex RPA primers were shorter (20–26 bp) than the recommended length of 30–35 bp, amplification of the $bla_{CTX-M}$, $bla_{OXA}$, and $bla_{SHV}$ genes was still accomplished and generated the product sizes from 296–593 bp. Longer primers with higher GC contents increase the probability of primer-dimer formation, while the optimal RPA amplicon size ranges from 100–200 bp. Longer products require a longer amplification time as well as an optimum ratio and concentration of primers [32].

Duplex and uniplex RPA products combined at a 2:4 (v/v) ratio provided specific detection of the three target genes on a lateral flow strip. Obvious blue lines for the specific RPA products corresponding to the $bla_{CTX-M}$, $bla_{SHV}$, and $bla_{OXA}$ genes appeared within 10 min. A false SHV result was demonstrated in the triplex RPA-LFA, which might have been due to the biotin primers self-pairing with the $bla_{SHV}$-tag primers and subsequently hybridizing with the immobilized complementary oligonucleotides on the strip. The LOD was 2.5 ng/25 μL with no cross-reaction against seven common human pathogens. The LOD of the RPA-LFA depends on several factors, including the amplification efficiency, the multiplex level of each primer, self-pairing among primers, and the different (biased) amplification efficiencies of the different primers. Primer design and extensive optimization of multiplex nucleic acid amplification incorporated with LFA assays are critical steps to avoid primer dimers and nonspecific results [31]. The coupling of powerful amplification methods with simple detection platforms, especially lateral flow strips, increases the feasibility of their application in point of care or field detection [33]. Of the various LFA formats, the STH-PAS is a lateral flow dipstick format that allows the detection of

**Table 4. Agreement analysis between RPA-LFA and PCR detection of $bla_{CTX-M}$, $bla_{OXA}$, and $bla_{SHV}$ (n = 140).**

| RPA-LFA | | PCR | | | | | | | | | | |
|---|---|---|---|---|---|---|---|---|---|---|---|---|
| | | CTX-M+ | CTX-M- | Totals | | OXA+ | OXA- | Totals | | SHV+ | SHV- | Totals |
| | CTX-M+ | 129 | 1 | 130 | OXA+ | 2 | 0 | 2 | SHV+ | 2 | 0 | 2 |
| | CTX-M- | 0 | 10 | 10 | OXA- | 0 | 138 | 138 | SHV- | 0 | 138 | 138 |
| Totals | | 129 | 11 | 140 | | 2 | 138 | 140 | | 2 | 138 | 140 |
| | Positive agreement: 99% CI 95% [95.8%-100%] | | | | Positive agreement: 100% CI 95% [15.8%-100%] | | | | Positive agreement: 100% CI 95% [15.8%-100%] | | | |
| | Negative agreement: 100% CI 95% [69.2%-100%] | | | | Negative agreement: 100% CI 95% [97.4%-100%] | | | | Negative agreement: 100% CI 95% [97.4%-100%] | | | |

multiple targets in a single reaction and easy visualization of the results in a short time by the naked eye. The optimum condition for STH-PAS detection depends on the stringency of the developing buffer, the ratio of products to developing buffer, and incubation time. For isothermal amplification, the reagent's instructions recommend using a modified dilution buffer and diluting the template five- to tenfold with TE buffer or water (TBA Co., Ltd.).

The sensitivity and specificity of our RPA-LFA for the detection all three common ESBL genes in pork and pig cecum samples were 99.2–100% and in almost perfect agreement ($\kappa$ = 0.949–1.00) with the PCR results. Multiplex PCR combined with STH-PAS has shown high sensitivity (93.3%) and specificity (99.1%) for the direct detection of carbapenemase genes from stool specimens within 2 h [34, 35]. The transmission of the *mcr*-1 gene and carbapenem-resistant *Enterobacteriaceae* has been determined using a PCR-STH-PAS dipstick and pulse-field gel electrophoresis [35]. However, no multiplex RPA combined with STH-PAS has been reported before. A combination of other common drug resistance genes, including *bla*-$_{TEM}$, plasmid-mediated AmpC beta lactamase genes, carbapenemase genes, and internal control lines, in one strip remains a challenge.

Overall, *E. coli* was found in 81% of the pork samples obtained from 100 retail markets in Bangkok, Thailand. This high level of contamination with *E. coli* is not only an indicator of poor hygiene but also indicates the distribution of antibiotic resistance in foods. Out of 98 *E. coli* isolates from pork that were tested, 93.9% were ESBL positive, and three isolates with carbapenem resistance were observed. Similarly, intestinal carriage of ESBL-positive *E. coli* in pigs was 96% in the tested slaughterhouse, which provides wholesale pork products throughout fresh markets in Bangkok and nearby provinces. High MDR profiles existed mostly in ESBL-positive *E. coli*, which implies a crisis in antibiotic usage. ESBL genes are located on plasmids, which are usually coexpressed with other plasmid-mediated drug resistance genes, such as quinolone, aminoglycosides, trimethoprim, and tetracyclines [36, 37]. Similar to a previous report, a high incidence of ESBLs with resistance to other drug classes was observed among food animals and fresh meats in Thailand [11, 38]. The presence of ESBL-MDR pathogens is a serious health problem due to the limited available drug regimens for treatment. Food producers, including farms and slaughterhouses, are important transmission sources of antibiotic resistance [39, 40].

The frequency of CTX-M was high in both pork (91.3%) and pig cecum samples (95.8%), with the majority being CTX-M-55. Only four subtypes (CTX-M-55, CTX-M-14, CTX-M-15, and CTX-M-27) were detected in the pig fecal samples from the slaughterhouse, while 14 CTX-M subtypes were detected in raw pork samples from shops that came from various slaughterhouses. A high prevalence of CTX-M-55 in pork has been reported previously in many countries, including Cambodia, Hong Kong, and Vietnam [41–43]. A total of 212 ESBL-producing *E. coli* isolated from healthy subjects in the community and swine of Lamphun Province, Thailand, revealed that 95.8% were positive for CTX-M, with the most common subtype being CTX-M-55, followed by TEM (60.9%) and SHV (2.4%) [44]. The emergence of CTX-M-55 has also been observed in *Salmonella* isolates from raw meat and food animals in China and Cambodia [42, 45]. However, the prevalence of ESBL subtypes varies in each geographical and epidemiological area. Antibiotic management is required in the whole production system, from farms to consumers. Moreover, the consumption of cooked meats is safer from food spoilage organisms and has a lower risk of antibiotic transmission than raw foods.

## Conclusion

The RPA-LFA (using STH-PAS) is a rapid and reliable tool for the detection of ESBL-producing *E. coli* in pork and fecal samples. RPA amplification can be performed using simple heating

equipment, and the results can be easily examined by the naked eye on a lateral flow strip. Thus, the assay is convenient for ESBL detection in food animals, patients, and environmental samples in low-resource settings.

## Author Contributions

**Conceptualization:** Nuntaree Chaichanawongsaroj.

**Formal analysis:** Sitthichai Kanokudom.

**Funding acquisition:** Nuntaree Chaichanawongsaroj.

**Methodology:** Sitthichai Kanokudom, Thachaporn Assawakongkarat.

**Supervision:** Yukihiro Akeda, Panan Ratthawongjirakul, Rungtip Chuanchuen, Nuntaree Chaichanawongsaroj.

**Validation:** Nuntaree Chaichanawongsaroj.

**Writing – original draft:** Sitthichai Kanokudom.

**Writing – review & editing:** Nuntaree Chaichanawongsaroj.

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
