## [Decision Letter · Decision Letter 0]

10 Feb 2021

PONE-D-20-40583

Rapid detection of extended spectrum β-lactamase producing Escherichia coli isolated from fresh pork meat and pig caecum samples using multiplex recombinase polymerase amplification and lateral flow strip analysis

PLOS ONE

Dear Dr. Chaichanawongsaroj,

Thank you for submitting your manuscript to PLOS ONE. After careful consideration, we feel that it has merit but does not fully meet PLOS ONE’s publication criteria as it currently stands. Therefore, we invite you to submit a revised version of the manuscript that addresses the points raised during the review process.

Please correct discrepancy in Table 3.

We look forward to receiving your revised manuscript.

Kind regards,

Iddya Karunasagar

Academic Editor

PLOS ONE

Additional Editor Comments:

In addition to what the reviewer has pointed out, it is noted than in Table 3, it is stated that n=140. Column 2 shows that 130 were PCR positive for blaCTX-M and column 3 shows the breakup of different combinations - either blaCTX-M alone or along with blaOXA and blaTEM and the total is 129. It should be 130 as per column 2. Why this discrepancy?

Journal Requirements:

Reviewers' comments:

Reviewer's Responses to Questions

**Comments to the Author**

1. Is the manuscript technically sound, and do the data support the conclusions?

Reviewer #1: Yes

2. Has the statistical analysis been performed appropriately and rigorously? 

Reviewer #1: Yes

3. Have the authors made all data underlying the findings in their manuscript fully available?

Reviewer #1: Yes

4. Is the manuscript presented in an intelligible fashion and written in standard English?

Reviewer #1: Yes

5. Review Comments to the Author

Reviewer #1: Manuscript no: PONE-D-20-40583 Manuscript title: Rapid detection of extended spectrum β-lactamase producing Escherichia coli isolated from fresh pork meat and pig caecum samples using multiplex recombinase polymerase amplification and lateral flow strip analysis

This manuscript describes development of multiplex RPA combined with LFA to detect the three most common ESBL genes (blaCTX-M, blaOXA, and blaSHV) and looked into their sensitivities and specificities. The assay developed in this study is convenient for ESBL detection in food animals, patients, and environmental samples in low resource settings. In this regard, the data presented in this manuscript is informative. I feel that this manuscript is worthy of publication in PLOS ONE after revision.

Following are the comments:

1. I would recommend the manuscript to be edited by native speaker of English prior to re-submission.

2. Fig 1. Spelling of “solution” need to be corrected

3. Pictures are not very clear. Could these photographs be improved at all?

6. PLOS authors have the option to publish the peer review history of their article (what does this mean?). If published, this will include your full peer review and any attached files.

Reviewer #1: **Yes: **Praveen Rai

---

## [Author Response · Author response to Decision Letter 0]

25 Feb 2021

Dear Editor-in-Chief, 

We thank reviewers for constructive comments. We have revised our manuscript and the followings are responses to comments of reviewers.

Response to Editor Comments:

• Please correct discrepancy in Table 3. It is noted than in Table 3, it is stated that n=140. Column 2 shows that 130 were PCR positive for blaCTX-M and column 3 shows the breakup of different combinations - either blaCTX-M alone or along with blaOXA and blaTEM and the total is 129. It should be 130 as per column 2. Why this discrepancy?

Response: The discrepancy in Table 3 is correct. The total number of blaCTX-M detected by PCR was 130, while RPA-LFA demonstrated only 129 of blaCTX-M. One PCR positive sample of blaCTX-M and blaTEM gave false negative by RPA-LFA. 

The clarification was incorporated in page 15, line 6-8: Of 130 blaCTX-M PCR-positive samples, a discrepancy was revealed in one of the blaCTX-M, blaTEM samples by RPA-LFA (Table 3).

 Addition correction has been done in Table 3, the number of blaCTM-M alone and blaCTX-M, blaTEM detected by RPA-LFA were 47 and 80, respectively.

Response to Reviewer #1:

1. I would recommend the manuscript to be edited by native speaker of English prior to re-submission.

Response: The revised manuscript has already English proof by AJE before re-submission. The corrections have been done as in “Manuscript with track changes”. 

2. Fig 1. Spelling of “solution” need to be corrected

Response: This typing error “solution” in Fig.1 has already corrected. 

3. Pictures are not very clear. Could these photographs be improved at all?

Response: This concern of photographs has already improved and check by Preflight Analysis and Conversion Engine (PACE) digital diagnostic tool to ensure that figures meet PLOS requirements. 

Additional correction.

1. The revised manuscript has been rechecked and edited to meets PLOS ONE’s style requirements. 

2. The detail of figure legend in Fig 1. has been incorporated in page8, line 17-21: “The C-PAS4 membrane strip has 3 position markers (red lines) and four imprinted tag oligonucleotides from the bottom to top lines (tag1-4; Left). The C-PAS4 membrane strip is inserted into a developing solution containing RPA amplicons (middle) for 5-10 min to allow the RPA amplicons to be trapped by the complementary tag sequences and generate visible blue lines (CTX, SHV, OXA) at the relevant positions, as indicated (right). ”.

---

## [Decision Letter · Decision Letter 1]

1 Mar 2021

Rapid detection of extended spectrum β-lactamase producing Escherichia coli isolated from fresh pork meat and pig cecum samples using multiplex recombinase polymerase amplification and lateral flow strip analysis

PONE-D-20-40583R1

Dear Dr. Chaichanawongsaroj,

We’re pleased to inform you that your manuscript has been judged scientifically suitable for publication and will be formally accepted for publication once it meets all outstanding technical requirements.

Kind regards,

Iddya Karunasagar

Academic Editor

PLOS ONE

Additional Editor Comments (optional):

All reviewer comments have been addressed,

Reviewers' comments:

Reviewer's Responses to Questions

**Comments to the Author**

1. If the authors have adequately addressed your comments raised in a previous round of review and you feel that this manuscript is now acceptable for publication, you may indicate that here to bypass the “Comments to the Author” section, enter your conflict of interest statement in the “Confidential to Editor” section, and submit your "Accept" recommendation.

Reviewer #1: All comments have been addressed

2. Is the manuscript technically sound, and do the data support the conclusions?

Reviewer #1: Yes

3. Has the statistical analysis been performed appropriately and rigorously? 

Reviewer #1: Yes

4. Have the authors made all data underlying the findings in their manuscript fully available?

Reviewer #1: Yes

5. Is the manuscript presented in an intelligible fashion and written in standard English?

Reviewer #1: Yes

6. Review Comments to the Author

Reviewer #1: Authors have satisfactorily response the comments of this reviewer.

Minor correction required in table 1: primer tag need to be changed from "3' " to "5'" end of the primers.

7. PLOS authors have the option to publish the peer review history of their article (what does this mean?). If published, this will include your full peer review and any attached files.

Reviewer #1: No

---

## [Editor Report · Acceptance letter]

4 Mar 2021

PONE-D-20-40583R1 

Rapid detection of extended spectrum β-lactamase producing *Escherichia coli* isolated from fresh pork meat and pig cecum samples using multiplex recombinase polymerase amplification and lateral flow strip analysis  

Dear Dr. Chaichanawongsaroj:

I'm pleased to inform you that your manuscript has been deemed suitable for publication in PLOS ONE. Congratulations! Your manuscript is now with our production department. 

Kind regards, 

on behalf of

Dr. Iddya Karunasagar 

Academic Editor

PLOS ONE